# Modulate stress distribution with bio-inspired irregular architected materials towards optimal tissue support

Yingqi Jia [1], Ke Liu [2] ✉ & Xiaojia Shelly Zhang [1,3,4] ✉

Natural materials typically exhibit irregular and non-periodic architectures, endowing them with compelling functionalities such as body protection, camouflage, and mechanical stress modulation. Among these functionalities, mechanical stress modulation is crucial for homeostasis regulation and tissue remodeling. Here, we uncover the relationship between stress modulation functionality and the irregularity of bio-inspired architected materials by a generative computational framework. This framework optimizes the spatial distribution of a limited set of basic building blocks and uses these blocks to assemble irregular materials with heterogeneous, disordered microstructures. Despite being irregular and non-periodic, the assembled materials display spatially varying properties that precisely modulate stress distribution towards target values in various control regions and load cases, echoing the robust stress modulation capability of natural materials. The performance of the generated irregular architected materials is experimentally validated with 3D printed physical samples − a good agreement with target stress distribution is observed. Owing to its capability to redirect loads while keeping a proper amount of stress to stimulate bone repair, we demonstrate the potential application of the stress-programmable architected materials as support in orthopedic femur restoration.

Natural materials usually possess irregular architecture at the microscale, featured by disorderedness, non-uniformity, and aperiodicity, which empowers functionally graded properties that are optimally tailored for overall functionalities such as homeostasis regulation[1], tissue remodeling[2], body protection[3], flight agility[4], and stress shielding[5]. Examples include wood[6], seashells[7], bone[5], spider silk[8], turtle shells[3], and bird feathers[4] (Fig. 1a). To reproduce the superior functionalities of the natural architected materials, their geometric features have been borrowed to design engineered materials, benefiting applications in electromagnetics[9–12], optics[13], and mechanical engineering[14–19]. In mechanical engineering, architected materials have been designed to demonstrate promising properties such as negative Poisson's ratio[20–22],

vibration control[23,24], mechanical clocking[25], and programmable nonlinear responses[26,27], among many others[28–34].

Unlike the highly irregular natural materials, most engineered architected materials are empirically designed by periodically tessellating well-known motifs inspired by crystalline solids and/or artistic patterns[35–39]. The study on irregular architected materials is still in its infancy, owing to the difficulty of effectively modeling their sophisticated three-dimensional (3D) geometries in a nearly infinite space. A few pioneering models have been proposed, including filtered random lattice[40–42], phase-separation induced spinodal foam[37,43,44], and a versatile, bio-inspired virtual growth process[45]. The last work builds single-scale, homogeneous disordered microstructures with diverse

[1]Department of Civil and Environmental Engineering, University of Illinois Urbana-Champaign, Urbana, IL 61801, USA. [2]Department of Advanced Manufacturing and Robotics, Peking University, Beijing 100871, China. [3]Department of Mechanical Science and Engineering, University of Illinois Urbana-Champaign, Urbana, IL 61801, USA. [4]National Center for Supercomputing Applications, Urbana, USA. ✉e-mail: liuke@pku.edu.cn; zhangxs@illinois.edu

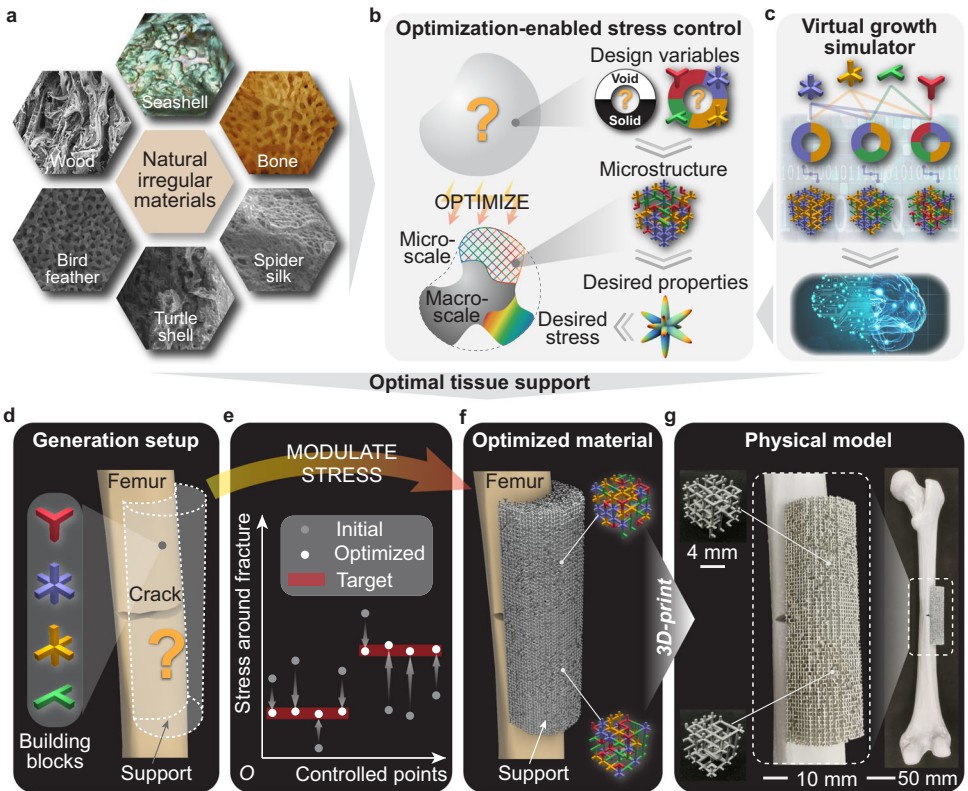

**Fig. 1 | Generation of irregular architected materials with optimization-enabled stress modulation and its potential applications. a** Natural irregular materials found in wood, seashells, bones, spider silk, turtle shells, and bird feathers. **b** A material property optimizer governs the design variables, microstructures, and material properties (directional elasticity), guiding the evolution of the bulk material at both macro and micro scales to achieve the desired stress distribution. **c** A virtual growth simulator produces diverse disordered microstructures based on the input frequency combinations of basic building blocks. A machine learning model further maps the frequency combinations of the building blocks to their corresponding material properties, which are fed back into the material property optimizer in (**c**). **d**–**g** The generation and fabrication of irregular architected materials used for orthopedic femur restoration after a fracture. **d** The generation setup including a fractured femur and a support to be filled with four types of basic building blocks. **e** The initial, optimized, and target stresses in the control region around the fracture, where the initial stress corresponds to a homogeneous distribution of building blocks. **f** The generated support made of optimized irregular materials, which are further composed of disordered microstructures illustrated in the two insets. **g** The 3D-printed samples at varying scales.

material properties by randomly nucleating basic building blocks. However, to develop superior functionalities as natural materials, it is necessary to introduce multi-scale, heterogeneous materials by embedding spatially varying and functionally graded microscopic features[46,47].

Here, we develop a generative computational framework that guides a virtual growth process to produce materials with heterogeneous disordered microstructures, optimized for overall functionalities such as stress manipulation. Our framework consists of two interconnected components: a material property optimizer (Fig. 1b) and a virtual growth simulator (Fig. 1c). In Fig. 1b, the material property optimizer assesses whether a sub-region of the bulk material should be solid or void and determines the optimal frequency combinations of basic building blocks. This information further defines the corresponding microstructure and the desired material properties, particularly directional elasticity, for that specific material sub-region. Consequently, the material property optimizer enables the evolution of the material at both the macro and micro scales, ensuring optimally distributed local properties that contribute to the desired global functionality, specifically stress modulation. In Fig. 1c, the virtual growth simulator[45] facilitates the seamless integration of disordered microstructures with varying homogenized material properties achieved through different frequency combinations of basic building blocks. Additionally, a machine learning model is employed to map these frequency combinations to their corresponding material properties, which are then fed back into the material property optimizer to

achieve the desired stress distribution. With the material property optimizer (Fig. 1b) and virtual growth simulator (Fig. 1c) working synergistically, our proposed framework generates functionally graded materials in a way analogous to how biological systems are built. The details of the proposed framework are introduced in the Methods section.

In comparison to related existing research[38,48] that primarily focuses on designing deterministic and nearly periodic architected materials, the present study introduces a generative computational framework for crafting stochastic and aperiodic materials. The incorporation of stochasticity and aperiodicity holds the potential to enhance failure resistance[49]. Furthermore, unlike the existing studies that predominantly investigate the performance (such as stiffness) of individual microstructures, our current study aims at modulating the stress distribution of functional material pieces comprising heterogeneous microstructures.

In this work, we harness the capabilities of the proposed generative computational framework to modulate the stress distribution across a spectrum of desired values, stress measures, control regions, and load cases. For example, we optimize a femur support with tailored irregular architected materials for promoting orthopedic restoration after a fracture (Fig. 1d–f and Supplementary Movie 1). The generated femur support precisely achieves the target stress distribution (Fig. 1e), preventing potential aseptic loosening and peri-prosthetic fracture[50], while keeping an appropriate amount of shear stress acting on the fractured region to stimulate bone regeneration[51,52]. A commercial

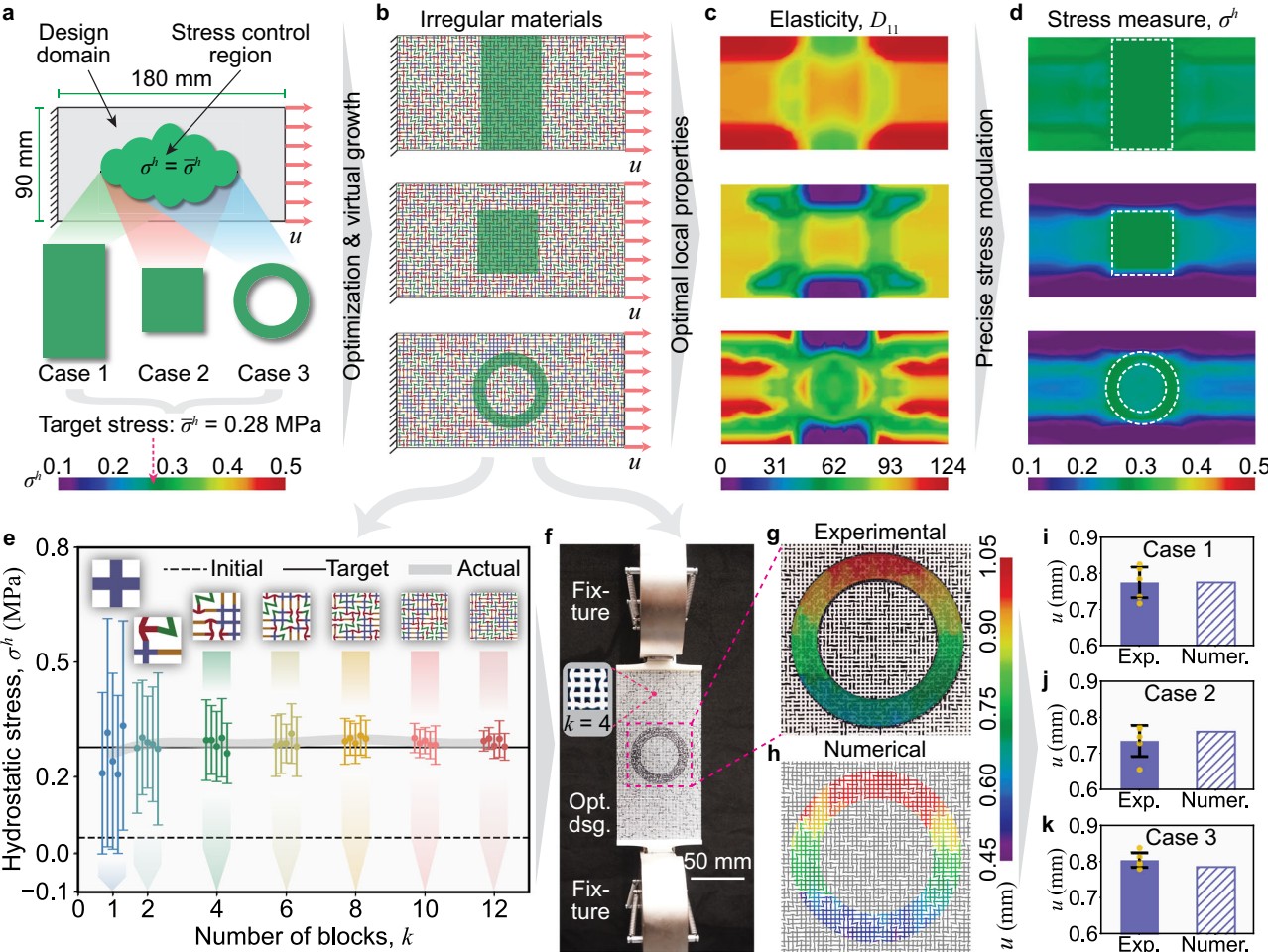

**Fig. 2 | Manipulating mechanical stress distribution in varied geometric regions. a** Design domain, boundary conditions, and three distinct stress control regions (rectangle, square, and ring-shape in Cases 1–3, respectively). The variable $u = 1.5$ mm represents displacement loading. Variables $\sigma^h$ and $\overline{\sigma}^h$ are actual and target hydrostatic stresses (in MPa), respectively. **b** Optimized irregular architected materials made of randomly yet optimally distributed microstructures. **c** Spatially varying distribution of properties, as represented by the $D_{11}$ elastic modulus (in MPa). **d** Precise stress manipulation (in MPa) is realized by the spatially varying material property. **e** Stress convergence study of the generated material in Case 3.

The variable $k$ represents the number of basic building blocks in one direction within one microstructure. Each error bar represents the distribution of hydrostatic stress of one specimen. The dots and the half-lengths of error bars indicate the mean values and the standard deviations, respectively. **f** Experimental setup for measuring the displacement field. **g,h** Displacements (in mm) in the loading direction obtained experimentally from the digital image correlation (DIC) and numerically from the finite element analysis (FEA), respectively. **i–k** Detailed comparisons of the average displacements within the control regions for the 3 cases. The measured values and error bars of experiments are also plotted.

masked-stereolithography (m-SLA) 3D printer is used to actualize our design (Fig. 1g) with reduced effort compared to other multi-scale structural design approaches[44]. This is because our generated materials are self-supporting ensembles of building blocks with a manufacturable minimum feature size.

## Results

### Manipulating mechanical stress distribution in varied geometric regions

We demonstrate our framework starting from generating irregular architected materials with programmed stress distribution within varying geometric regions. The design domain and boundary conditions are shown in Fig. 2a, where we consider three stress control regions with different geometries: rectangle in Case 1, square in Case 2, and ring-shape in Case 3. The objective is to control the hydrostatic stress, defined by $\sigma^h = (\sigma_1 + \sigma_2 + \sigma_3)/3$, towards a target value $\overline{\sigma}^h = 0.28$ MPa, where $\sigma_1$, $\sigma_2$, and $\sigma_3$ are the three principal stresses[53]. We define four types of basic building blocks: cross, arrow, corner, and line (see Supplementary Fig. 1), which compose the irregular architected material. We optimize the frequency combination of the four building

blocks over different locations within the material, and grow the irregular architected materials accordingly. As shown in Fig. 2b, these irregular architected materials are made of randomly yet optimally distributed microstructures, which are seamlessly connected to ensure material integrity. Although the microstructures display a similar appearance across the entire piece, they show the spatially varying distribution of properties, as represented by the $D_{11}$ elastic modulus plotted in Fig. 2c. Such spatially varying material property is optimized to manipulate the stress distribution within the material to achieve desired values (Fig. 2d). Here, we note that this excellent stress modulation effect is not confined to specific design setups, and the current setup (Fig. 2a) is solely for demonstration purposes. Based on Supplementary Fig. 6 and Supplementary Table 1, the satisfactory stress modulation effect can be obtained under varying stress measures, target values, and applied displacements.

To examine the accuracy and reproducibility of the stress control effect of irregular architected materials, we select Case 3 as a representative material to perform a stress convergence study. As shown in Fig. 2e, we denote the number of basic building blocks in one direction within one microstructure as the variable $k$. We investigate how the

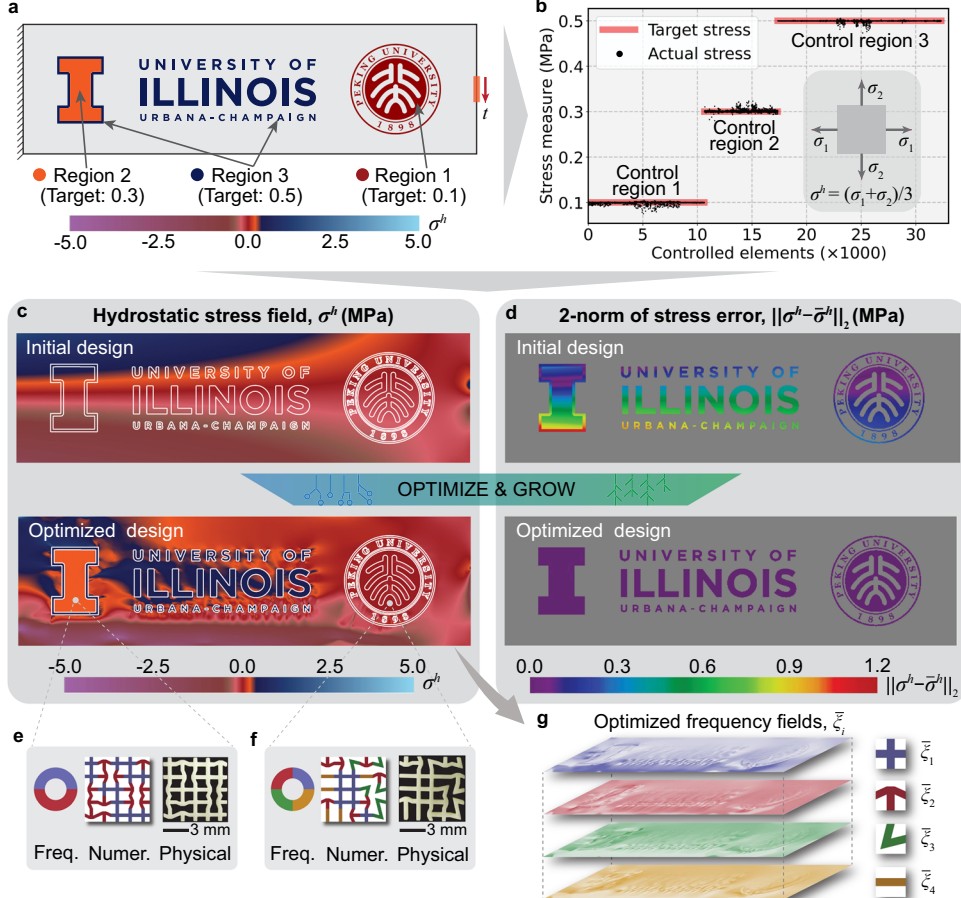

**Fig. 3 | Simultaneous mechanical stress modulation in multiple complex regions with distinct target values. a** Design domain, boundary conditions, stress control regions (divided according to colors), and target stress values ($\overline{\sigma}^h$, in MPa). The variables $\sigma^h$ and $t = 1$ MPa are the hydrostatic stress (in MPa) and the applied traction, respectively. **b** Comparison between the actual and target hydrostatic stresses after optimization. The inset shows the computation of the hydrostatic stress, $\sigma^h$, expressed in principal stresses, $\sigma_1$ and $\sigma_2$, under the plane stress assumption. **c** Hydrostatic stress field, $\sigma^h$, of the initial and optimized designs. **d** 2-norm errors of hydrostatic stress fields, $||\sigma^h - \overline{\sigma}^h||_2$, of the initial and optimized designs. **e,f** Two insets illustrating disordered microstructures of the optimized architected materials at two different locations of the sample. Each inset exhibits one frequency combination of a disordered microstructure and corresponding numerical and physical microstructural samples. **g** Optimized distribution of frequency combinations, $\overline{\xi}_1 - \overline{\xi}_4$, for each type of basic building block.

hydrostatic stress ($\sigma^h$) in the control region varies with different $k$ values ($k = 1, 2, 4, 6, 8, 10, 12$). The larger the $k$ value, the smaller the microstructural feature size. Considering the randomness of irregular architected materials, we generate 5 specimens for each $k$ value based on the same spatial distribution of frequency combinations. For each specimen, we perform the finite element analysis (FEA) at the microstructural level and obtain the mean value ($\mu$) and the standard deviation ($s$) of the computed hydrostatic stress in the control region (see Methods section). In Fig. 2e, the mean value ($\mu$) converges to the target, and the standard deviation ($s$) decreases as $k$ increases, indicating the improving accuracy of the stress control effect. This improvement results from the separation of the length scales between the bulk material and the microstructures for large $k$ values. Comparing $\mu$ and $s$ of the 5 specimens with the same $k$, we observe that they stay close to each other for $k > 1$, demonstrating reliable reproducibility of the stress control effects. Similar conclusions for Cases 1 and 2 are drawn in Supplementary Section 5.

To examine the computed architected materials, we use 3D printing (m-SLA) to manufacture five replicates of each sample shown in Fig. 2b with $k = 4$, and conduct experiments with setups shown in Fig. 2f (see Methods section). Figure 2g and h show the displacement fields in the loading direction of one experimental sample (corresponding to Case 3 in Fig. 2a) via digital image correlation (DIC) and

one numerical sample via FEA − consistent displacement fields are observed. Detailed comparisons are presented in Fig. 2i–k on the average displacements within the control regions for the 3 cases, respectively. The discrepancy between the experimental and numerical results is negligible, demonstrating reliable stress modulation of the optimized irregular architected materials, given that the stress error is proportional to the displacement error considering linear elasticity.

## Simultaneous mechanical stress modulation in multiple complex regions with distinct target values

We further elucidate the feasibility of leveraging irregular architected materials to simultaneously achieve distinct target stress values across multiple geometrically complex control regions. The design domain, boundary conditions, and stress modulation regions are shown in Fig. 3a. We divide these complex patterns into three control regions by their colors. The objective is to simultaneously modulate the hydrostatic stresses ($\sigma^h$) in these regions to different target values, $\overline{\sigma}^h = 0.1$, 0.3, and 0.5 MPa for control regions 1−3, respectively. After optimization, the converged (numerical) hydrostatic stresses approach the target values, as shown in Fig. 3b.

For better visualization, we plot the hydrostatic stress field, $\sigma^h$, for both the initial (homogeneous distribution of basic building blocks)

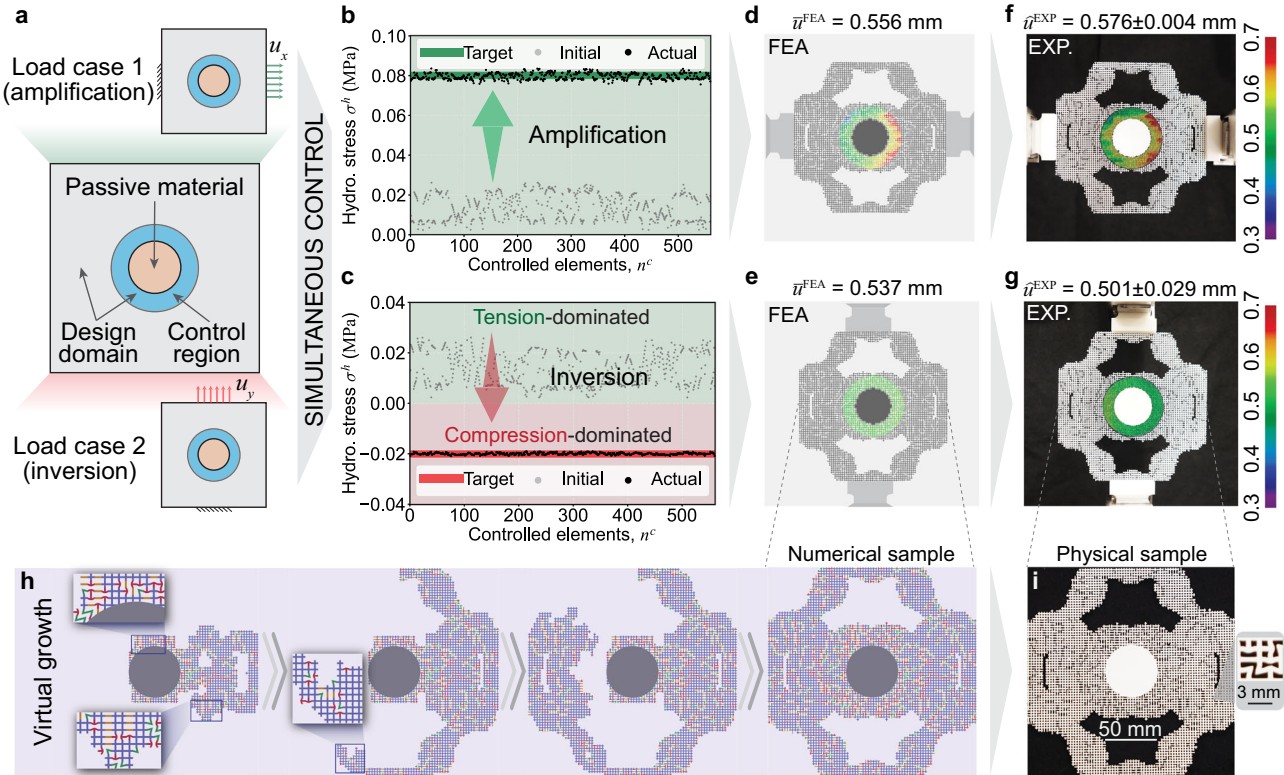

**Fig. 4 | Lightweight irregular architected materials optimized for multi-functional stress modulation. a** A squared design domain containing a passive material in the center subjected to two different load cases with applied displacements, $u_x = 1.0$ mm and $u_y = 1.0$ mm, respectively. The stress control region is around the passive material. **b, c** The stress amplification and inversion effects achieved in two load cases, respectively. **d, e** The displacement fields in the loading direction in two load cases obtained from the finite element analysis (FEA), respectively. **f, g** Corresponding displacement fields measured in experiments. **h** The virtual growing process of the optimized material piece. **i** A physical sample manufactured via 3D printing (m-SLA).

and optimized materials in Fig. 3c. As expected, by optimizing the heterogeneous distribution of basic building blocks, the hydrostatic stress within the irregular architected material is accurately modulated to meet the target values (Fig. 3c). We also calculate the 2-norm error of the hydrostatic stress field, $||\sigma^h - \bar{\sigma}^h||_2$, for both the initial and the optimized materials (Fig. 3d). Figure 3e–f illustrate the disordered microstructures of the optimized architected materials at two different locations of the sample. The optimized positions and orientations of basic building blocks in these irregular architected materials can redirect the load path and then contribute to the stress modulation. Figure 3g shows the optimized heterogeneous distribution of frequency combinations for each type of basic building block. Here, we reiterate that the proposed generative computational framework is not limited to specific target stress values. Supplementary Fig. 7 showcases the stress modulation effects for various orders of target stress, and the actual stress aligns closely with the target ones, indicating the robust performance of the framework.

### Lightweight architected materials for multifunctional stress modulation

We now focus on lightweight and multifunctional architected materials with optimized global material layout (built upon density-based topology optimization[26,27,54]) and local frequency combination of microstructures (Supplementary Movie 2). Figure 4a shows the design domain, boundary conditions, and the stress control region. Our objective is to control the hydrostatic stress in the same material to achieve two different stress modulation effects − stress amplification and inversion − in two load cases. The optimized material achieves both effects simultaneously, as shown in Fig. 4b, c. These figures depict the target, initial, and optimized hydrostatic stresses in the control region. Under Load Case 1 (Fig. 4b), the hydrostatic stress increases eightfold from around 0.01 to 0.08 MPa, demonstrating a significant stress amplification effect. In contrast, under Load Case 2 (Fig. 4c), the hydrostatic stress inverts from positive (tension-dominated) to negative (compression-dominated) signs within the control region even if under the tensile loading.

To validate such a multifunctional modulation effect, we perform the experiments (Fig. 4f and g) and compare the results side by side with the numerical ones obtained via FEA (Fig. 4d and e). Specifically, we examine the average displacements in the loading direction within the control region under two load cases. FEA predicts the average displacement as $\bar{u}_x^{FEA} = 0.556$ mm and $\bar{u}_y^{FEA} = 0.537$ mm in the two load cases, respectively. Experiments predicts the corresponding average displacements as $\hat{u}_x^{DIC} = 0.576 \pm 0.004$ mm and $\hat{u}_y^{DIC} = 0.501 \pm 0.029$ mm (further averaged over three replicates). Based on these measurements, we compute the errors as 3.5% and 7.1% for the two loading cases, respectively, and conclude the multifunctional stress modulation effect is reliable.

We also present the virtual growing process for generating optimized materials with $k = 1$ in Fig. 4h. The basic building blocks grow randomly yet optimally and seamlessly, mimicking the growth of natural materials. Figure 4i shows the physical sample manufactured via 3D printing (m-SLA). Such fabrication is effortless because the generated samples are self-supporting ensembles of basic building blocks with manufacturable minimum feature sizes (larger than 400 $\mu$m for m-SLA). Most importantly, this example demonstrates the possibility of simultaneously optimizing the global-level material layout and the local-level frequency combination of microstructures. Such simultaneous optimization enables versatile yet lightweight materials promising for biological applications[55].

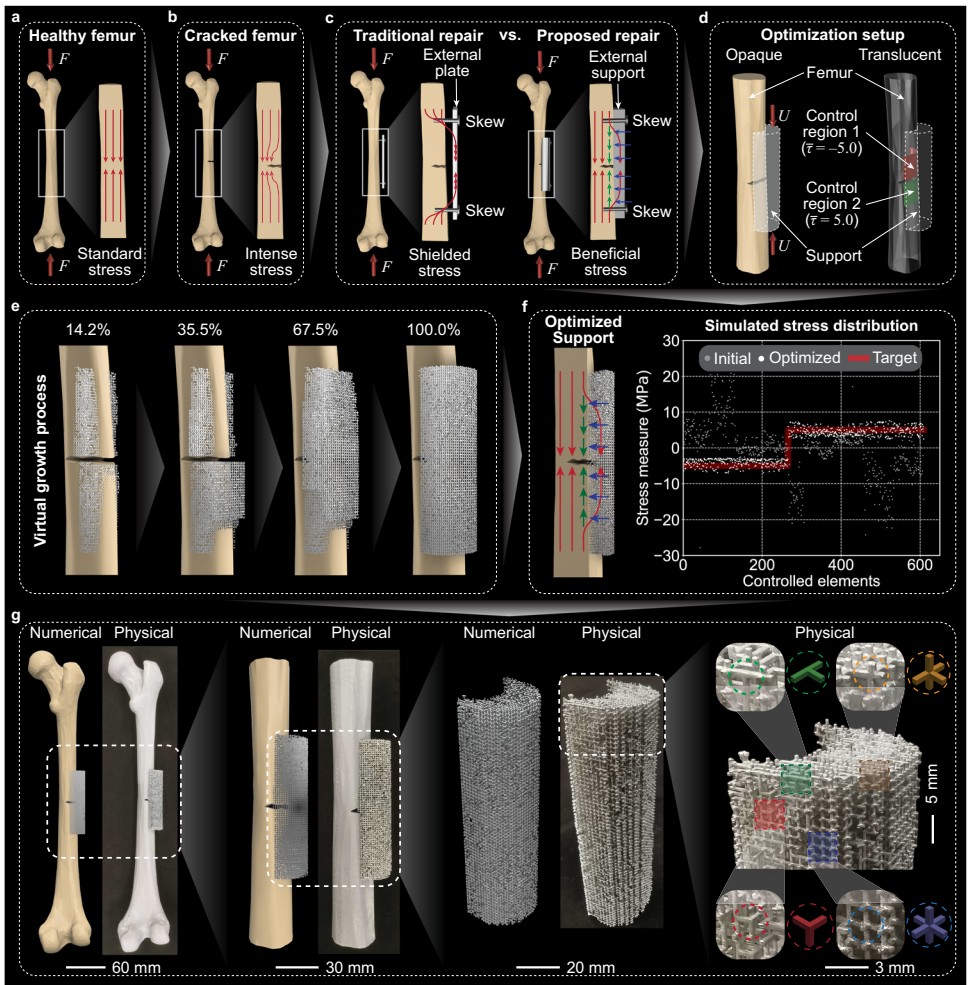

**Fig. 5 | Potential application of the irregular architected materials for orthopedic femur restoration. a** A healthy femur efficiently distributes compressive forces, resulting in relatively uniform compressive stress distribution (red arrows in the inset). **b** A fractured femur experiences concentrated compressive stress around the crack tip. **c** Comparison between the traditional and proposed methods for orthopedic femur restoration. The traditional method induces stress shielding, while the proposed method stimulates growth with shear stress (green arrows) under the transmitted pressure (blue arrows). **d** Simplified model of the proposed femur repair scheme with the applied displacement loading of $U$. Unit of target stresses, $\bar{\tau}$: MPa. **e** The virtual growth process of the optimized 3D irregular architected materials. **f** Illustrative (left) and simulated (right) stress distribution. In the right panel, the $x$-axis represents the finite elements in the stress control regions, and the $y$-axis represents the values of the selected stress measure (shear stress here). **g** Comparison between the numerical and physical samples of optimized architected materials at varying scales.

## Potential application to orthopedic femur restoration

Finally, we present a biomedical application of orthopedic femur restoration using optimized irregular architected materials demonstrating versatile mechanical stress modulation. Such modulation promotes the regeneration of fractured femurs with precise shear stress stimulation, avoiding stress shielding caused by traditional metal plates or intramedullary rods[56,57]. As illustrated in Fig. 5a, a healthy femur can efficiently transmit forces, $F$, applied at its two ends and yields a relatively uniform compressive stress field (red arrows in the inset). Unfortunately, femur fractures are prevalent, especially among elder individuals, and result in stress concentration at the crack tip, increasing the risk of further fracture propagation[58] (Fig. 5b). Traditional methods of repairing a fractured femur involve attaching a stiffer external plate around the fracture with screws (left part in Fig. 5c). Due to the stiffness difference between the femur and the external plate, stress shielding occurs, which further induces aseptic loosening, chronic pain, and periprosthetic fracture[56,57]. To address this issue, one potential solution is to utilize support made of architected materials (right part in Fig. 5c). This support system partially transmits compressive stress and then initiates shear stress (green arrows) when subjected to transverse pressure (blue arrows) between the femur and the support. Consequently, this shear stress induces relative micromotion perpendicular to the pre-existing fracture, thereby stimulating femur regeneration[59]. In this work, for illustrative purposes, we designate the target shear stresses as $\bar{\tau} = -5.0$ and $\bar{\tau} = 5.0$ MPa for control regions 1 and 2 (the translucent view in Fig. 5d), respectively. These specified target values are intentionally well below the shear strength range of human femurs (51.6 MPa–65.3 MPa for cortical bone[60]) to mitigate the risk of inducing additional fractures. Notably, these target stress values correspond to a relative micromotion of 0.3 mm of the femur around the fracture (see Supplementary Fig. 8), falling within the suggested range of [0.2, 1.0] mm for femur restoration[59]. A crucial aspect that must be addressed is the precise modulation of the shear stress to the desired level.

The specific problem is shown in Fig. 5d. Simplifying our femur model, we treat it as an isotropic passive material with Young's modulus of $E = 5.0$ GPa. This selection is informed by the predominant cortical bone composition in the femur's diaphysis, where Young's modulus is approximately 16.7 GPa[57]. In addition, the homogenized Young's modulus of the femur should be further reduced to account for the marrow cavity within the femur whose Young's modulus ranges

from 0.25 to 24.7 kPa[61]. Furthermore, the load-bearing defect (fracture) is directly reflected in the femur geometries (see Supplementary Fig. 8). Here, we note that the isotropy assumption yields results comparable to those obtained with orthotropic materials[62], and the proposed framework is applicable for a more intricate femur modeling with additional computational cost.

The objective in this example is to control the shear stress on the femur surface and around the fractured region to target values (Fig. 5d) that are assumed to stimulate tissue regeneration. To accomplish this goal, we utilize the proposed framework to optimize the heterogenous distribution of microstructures and then grow the irregular architected materials (Fig. 5e and Supplementary Movie 1). To visualize the stress modulation effect, Fig. 5f presents illustrative (left) and simulated (right) stress distributions. In the simulated stress distribution, the $x$-axis represents the finite elements in the stress control regions, while the $y$-axis represents the values of the selected stress measure (shear stress in this case). It is important to note that the initial stress distribution among the controlled elements is highly nonuniform due to the complex geometry of the femur (i.e., varying cross-sections and embedded fracture), despite the initial homogeneous distribution of building blocks within the support domain. However, after optimization, the actual stresses uniformly reach the target values. To validate the manufacturability of these materials, we use approachable 3D printing (m-SLA) to fabricate the generated architected materials. The fabricated samples are compared with the numerical ones side-by-side (Fig. 5g) − each printed building block is as small as $1.5 \times 1.5 \times 1.5$ mm$^3$. Such promising manufacturability benefits from the generated materials' self-supporting feature and the pre-defined feature size of the basic building blocks.

## Discussion

In this study, we uncover the fundamental relationship between the stress modulation functionality and the irregularity of bio-inspired architected materials, which are created by a generative computational framework. This framework employs a material property optimizer to enable the overall stress modulation functionality and a virtual growth simulator to seamlessly create irregular architected materials that conform with the desired property distribution within a piece of material. We examine the stress modulation functionality of generated materials for various desired values, stress measures, control regions, and load cases − a good agreement with target stress distribution is observed, both numerically via FEA, and experimentally via DIC. Such stress modulation functionality can be promising for biomedical applications. For instance, we demonstrate the potential application of the irregular architected materials as a support structure in orthopedic femur restoration that is capable of redirecting the load while stimulating tissue regeneration. In addition, the generated architected materials are self-supporting ensembles of basic building blocks with a manufacturable minimum feature size, free of any thin and dangling parts, and can be manufactured by approachable 3D printing technologies. We remark that the proposed framework is ready for extension to handle complex scenarios involving multiphysics and nonlinear responses. Such extension has the potential to benefit various engineering applications, including biomedical devices, bio-mimetic robots, and lightweight space structures.

Moving forward, we anticipate incorporating more biological considerations into the framework and conducting in-vivo tests to validate the designed support for orthopedic femur restoration. To achieve this goal, we plan to employ a more representative femur model reflecting its original porous structures and spatially varying material properties. In addition, we need to incorporate biological design variables[63] including the porosity, pore size, and the specific surface area of the support and the biological constraints, including the minimum stiffness and fracture resistance[58] of the femur. We also need to ensure the biocompatibility of the support material[64].

Furthermore, during femur restoration, as the bone consolidates and the callus stiffness increases, enlarging the target shear stress may be necessary to maintain the same level of micromovement of femur fragments around the fracture. To accommodate varying target stress levels, we can reformulate the design task into a simultaneous control problem akin to Fig. 4. Consequently, the generated support can modulate stress to different levels as the femur progresses through different healing states.

Moreover, in the context of orthopedic femur restoration, our assumption entails a perfect bonding interface between the femur and the support. It is crucial to note that in real in-vivo applications, aligning artificial materials with biological tissues demands the expertise and careful consideration of the operator. Additionally, there is a possibility of local stress concentration along the interface between the femur and the support composed of artificial microstructures. Despite the potential for stress concentration, the proposed generative computational framework primarily modulates the homogenized stress distribution rather than focusing on local stress patterns. This approach allows for the control of the global relative displacement of the femur perpendicular to the fracture, as illustrated in Supplementary Fig. 8, with the ultimate goal of stimulating tissue regeneration. We also note that the efficacy of such stimulus becomes more reliable, characterized by fewer variations, when employing a greater number of building blocks in each microstructure (Fig. 2e and Supplementary Fig. 10).

Beyond biological considerations, the stress modulation performance can be further enhanced by exploring alternative building blocks. In our current study, we concentrate on investigating a specific set of building blocks (in 2D and 3D, respectively) to modulate stress distribution. Referring to Figs. 4, 6 in ref. 45, the use of alternative building blocks with different geometries can significantly impact the material property space and, consequently, influence stress modulation effects. Additionally, due to the anisotropy of material properties in microstructures, expanding the material property space by allowing free elongation and rotation (as opposed to restricting it to multiples of $\pi/2$) is a potential improvement.

## Methods

The proposed generative computational framework (Fig. 6) consists of four steps: creating a material database, training a machine learning model, performing macroscopic topology optimization, and applying a bio-inspired virtual growth simulator. Below, we present the overall framework along with its four components, as well as the associated manufacturing and experimental setups.

### Overall framework

To generate the irregular architected materials, Fig. 6 provides a detailed exposition of the proposed generative computational framework. As shown in Fig. 6a, this framework begins by creating a discrete material database containing the prescribed building blocks, frequency combinations, and the generated microstructures. Following numerical homogenization[65], the material properties of the microstructures are derived, and these properties are then correlated with the microstructures' frequency combinations. To establish a continuous relationship between the frequency combination and the material properties, a machine learning model is trained (Fig. 6b) to predict the independent components of the stiffness matrix based on the input frequency combination. Subsequently, macroscopic topology optimization is performed (Fig. 6c) to optimize the design variables defined on the finite elements within the design domain to meet the target. During this optimization process, a density variable is defined to describe whether a finite element is filled with a microstructure (solid) or not (void). Additionally, frequency variables are defined to characterize the frequency combination of building blocks if the finite element is filled with a microstructure. After defining these

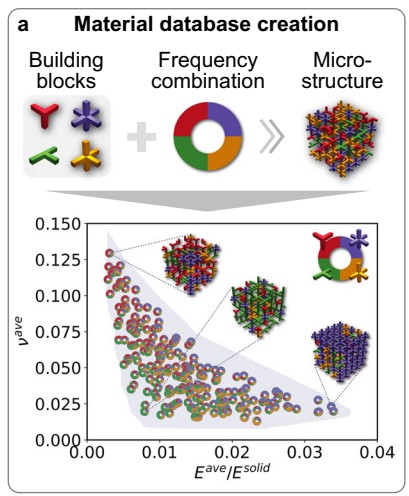
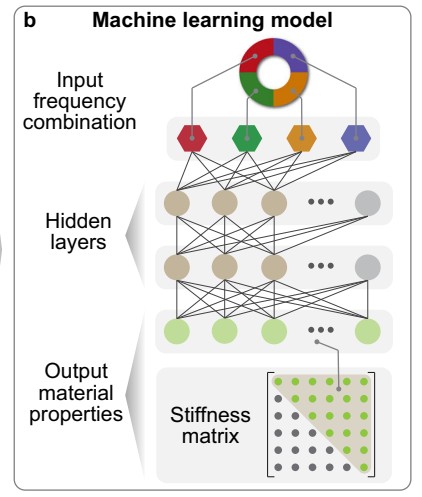
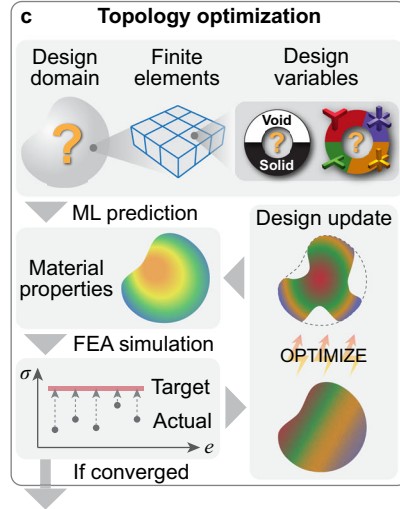

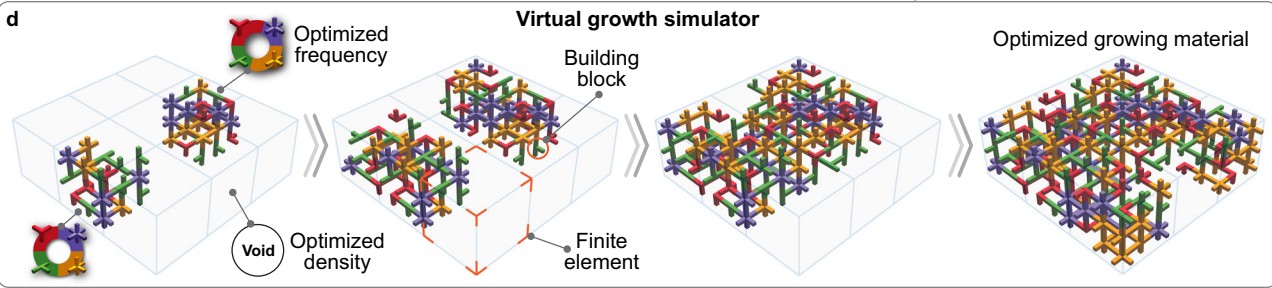

**Fig. 6 | Proposed generative computational framework. a** Material database creation. The bottom panel is the material property space, and the details are explained in Supplementary Section 1.4. **b** Machine learning (ML) model for predicting the material properties based on the input frequency combination.

**c** Macroscopic topology optimization that determines the optimal design variables to modulate the stress distribution to the target. **d** Virtual growth simulator for generating the irregular architected materials based on the optimized design variables.

design variables, the machine learning model from Fig. 6b is utilized to predict the spatial distribution of material properties. Subsequently, FEA is conducted to derive the stress distribution. Guided by the gradient information, the design variables undergo iterative updates by the optimizer until the actual stress distribution converges to the target. Finally, based on the optimized density and frequency variables, we apply the virtual growth algorithm to grow the irregular architected materials capable of modulating the stress distribution (Fig. 6d).

Here, we remark that the original virtual growth algorithm in[45] mainly yields materials with one intuitively specified frequency combination (i.e., one finite element and one microstructure). In the current study, we generalize this algorithm to account for spatially varying and optimized frequency combinations and densities (solid or void) within different finite elements. In the generalized algorithm (Fig. 6d), individual finite elements are assigned unique frequency combinations, and all building blocks within a finite element adhere to the corresponding frequency combination of that finite element. Consequently, the generated materials consist of heterogeneous microstructures.

### Creating a material database
To create the material database, we uniformly sample 200 frequency combinations of four basic building blocks, $\bar{\xi}_1 - \bar{\xi}_4$, on the hyperplane $\sum_{i=1}^{4} \bar{\xi}_i = 1$, for both two-dimensional (2D) and three-dimensional (3D) cases. Using the virtual growth algorithm[45], we then generate 100 specimens of irregular architected materials for each frequency combination, resulting in 20,000 square specimens in 2D and 20,000 cubic specimens in 3D. Each 2D specimen contains $40 \times 40$ building blocks, and each 3D specimen contains $10 \times 10 \times 10$ building blocks. To obtain the homogenized elasticity tensor (a $3 \times 3$ matrix in 2D and a $6 \times 6$ matrix in 3D in matrix notation[66]) of the irregular

architected material, we utilize a numerical homogenization approach in[65]. Specifically, we use first-order quadrilateral elements in 2D and frame elements in 3D in FEA to compute the global stiffness matrix. We then compute the average homogenized elasticity tensor for every 100 specimens corresponding to the same frequency combination. This process establishes the constitutive relationships between the frequency combinations and the homogenized elasticity tensors of disordered microstructures, resulting in 200 pairs in 2D and 200 pairs in 3D. See more details in Supplementary Section 1.

### Training a machine learning model
To establish a continuous and differentiable relationship between the frequency combination and the homogenized elasticity tensor of disordered microstructures, we train a fully connected neural network to capture such complex information. This neural network comprises one input layer with 4 nodes corresponding to the frequencies of 4 basic building blocks ($\bar{\xi}_1 - \bar{\xi}_4$), two hidden layers with 512 and 256 nodes, respectively, and one output layer with 6 nodes in 2D and 21 nodes in 3D corresponding to the maximum numbers of independent entries in the homogenized elasticity tensor. We employ the rectified linear activation unit (ReLU) as the activation function, the mean squared error (MSE) as the loss function, and the Adam algorithm[67] as the optimizer. A total number of 200 frequency–property pairs in the previous material database are randomly classified into training data (160 pairs) and testing data (40 pairs) for both 2D and 3D cases, respectively. To optimize the neural network's parameters, we minimize the loss function. We train the neural networks for 1000 epochs by using an adaptive learning rate, $10^{-2}$ for the first 200 epochs, $10^{-3}$ for epochs 201–500, and $10^{-4}$ for the remaining epochs. After the training, the neural network performs well in predicting the homogenized elastic properties, as evidenced by the small MSEs of 0.134 MPa in 2D

and 0.296 MPa in 3D, in comparison to the corresponding mean values of 2.687 MPa in 2D and 7.344 MPa in 3D. See more details in Supplementary Section 2.

## Performing macroscopic topology optimization

To optimize the layout and the frequency combination of disordered microstructures, we propose a macroscopic topology optimization framework consisting of the following procedures. First, we parameterize the microstructural layout with a density field, $\overline{\rho} \in [0,1]$, and the microstructural frequency combination with frequency fields, $\overline{\xi}_i \in [0,1]$ for $i = 1, 2, ..., N_b$, where $N_b = 4$ is the number of types of basic building blocks. The density field $\overline{\rho} = 1$ represents the presence of microstructures, while $\overline{\rho} = 0$ corresponds to voids. The frequency field $\overline{\xi}_i$ denotes the frequency of the $i$-th basic building block. Second, we interpolate the elasticity tensor, $\mathbf{D}(\overline{\rho}, \overline{\xi}_1, \overline{\xi}_2, \ldots, \overline{\xi}_{N_b})$, as a function of the density and frequency fields to compute the stress response in FEA. Next, we cast an objective function, $J$, to represent the stress modulation error for varying target values, stress measures, control regions, and load cases. Finally, we formulate the topology optimization problem to find the spatially varying $\overline{\rho}$ and $\overline{\xi}_i$, minimizing $J$, subjected to a material volume constraint. We implement this optimization formulation with the Python programming language[68] and the FEniCSx package[69]. See more details in Supplementary Section 3.

## Applying a bio-inspired virtual growth simulator

Based on optimized density and frequency fields, we use the virtual growth simulator built upon[45] to generate the microscopic irregular architected materials. Specifically, we propose a two-mesh-projection scheme (see Supplementary Section 4.2) to project the optimized density ($\overline{\rho}$) and frequency ($\overline{\xi}_i$ for $i = 1$ to $N_b$) fields obtained from topology optimization onto a structured mesh containing square grids in 2D and cubic grids in 3D, and each grid corresponds to one disordered microstructure. The projected density field determines whether a grid should be filled with microstructure or not. If filled, the projected frequency fields further determine the optimal frequency combination of this grid and guide the virtual growth simulator to grow basic building blocks. During the virtual growing process, each newly filled building block needs to satisfy the prescribed frequency combination and adjacency rules associated with its neighboring blocks. Eventually, we create an optimal and seamless integration of disordered microstructures. Notably, the virtual growth simulator used in this work extends the original virtual growth program[45] in two aspects. First, the original program focuses on a homogeneous distribution of basic building blocks, while our simulator features a heterogeneous distribution of building blocks. Second, the original program relies on intuition-based input frequency combinations, while the current approach follows the optimized density and frequency fields obtained from macroscopic topology optimization. These two improvements enable optimal microstructural layouts and customized local material properties, which are crucial for precise stress modulation. See more details in Supplementary Section 4.

## Manufacturing by 3D printing

The physical samples of the optimized irregular architected materials are 3D printed using the m-SLA technology. Specifically, we use an Elegoo Saturn 2 m-SLA 3D printer with Elegoo water-washable resin materials. This printer features a layer-by-layer curing process by using ultraviolet light shining on a resin vat. The allowable printing volume is $219 \times 123 \times 250$ mm$^3$, and the layer height ranges from $10$–$200\,\mu m$. Based on our printing tests, the minimum printable feature size is $400\,\mu m$. The 3D printing involves the following steps: generating geometric models using an in-house Python code, converting geometric models to STereoLithography (STL) files with PyVista software[70], converting STL files to Color Dependent Plot Style (CTB) files with Chitubox software, printing CTB files with the Elegoo printer,

washing printed samples with Original Prusa Curing and Washing Machine (CW1S), drying washed samples with tissue papers, and curing dried samples with CW1S again. The majority of specimens are printed in one piece, except for the full-scale femur due to the limited printing volume. We print this full-scale femur in two parts and bond them with super glue (Loctite, Henkel Corporation). Finally, we apply metallic paint (Krylon Fusion All-In-One, Krylon Products Group) on printed architected materials for better visualization.

## Mechanical characterization

We perform all experiments using an Instron 68TM-30 universal testing machine. For material characterization, we manufacture five dogbone-shaped specimens and perform uniaxial tensile tests with a 5 mm/min loading rate via a 300 kN load cell. We obtain the average Young's modulus and the average Poisson's ratio as 1162.51 MPa and 0.40, respectively. For testing the stress modulation effect (via displacement fields), we employ the same setup of uniaxial loading for all irregular architected materials. To track the displacement field, we use a marker pen to draw dispersed markers on the stress control region. We then record videos of the marker movement with a camera (Sony Alpha 7R III, Sony Corporation) and analyze the displacement field using the DIC technique by Ncorr software[71].

## Data availability

The datasets generated in this study are provided in the Source Data file. Source data are provided within this paper.

## Code availability

The computer code that supports the findings of this study has been deposited in the GitHub repository at https://github.com/jiayingqi/Heterogeneous-Virtual-Growth under accession code https://doi.org/10.5281/zenodo.10963129. The reader can download and execute the code with guidance from the README document.

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

## Acknowledgements

Authors X.S.Z. and Y.J. acknowledge the support from the David C. Crawford Faculty Scholar Award from the Department of Civil and Environmental Engineering and Grainger College of Engineering at the University of Illinois. The information provided in this paper is the sole opinion of the authors and does not necessarily reflect the view of the sponsoring agencies.

## Author contributions

X.S.Z. and K.L. conceived the idea and supervised the research project; Y.J., K.L., and X.S.Z. designed the generative computational framework, analyzed the data, and wrote the manuscript; Y.J. developed the computer code, implemented the numerical simulation, fabricated the samples, and performed the experiments; X.S.Z. secured the funding support and provided the resources. All authors contributed to the manuscript revision and approved the submitted version.

## Competing interests

The authors declare no competing interests.
