## [Peer Review File · Nature Communications]

Reviewers' Comments:

Reviewer #1:

Remarks to the Author:

The paper investigates the relationship between stress modulation and the irregularity of bio-inspired architected materials. It introduces a computational framework for optimizing the spatial distribution of basic building blocks to create irregular architected materials with heterogeneous disordered microstructures. The work includes a practical demonstration of this concept through 3D printed samples and highlights potential applications in orthopedic femur restoration. Overall, the reviewer is positive about the work - especially, the novel aspects regarding modulating stress distribution of large-scale irregular architected materials.

My comments are listed below:

(1) It is unclear what criteria guided the author's selection of the four fundamental 3D building blocks. The paper would benefit from a discussion exploring alternative building blocks that could potentially enhance performance.

(2) As shown in Figure 5f, the meaning of the 'controlled element' on the x-axis requires clarification. Furthermore, it is not clear to the reviewer why a significant stress disparity is observed among different controlled elements, while the initial state suggests a homogeneous distribution of building blocks.

(3) The proposed framework integrates a machine learning model, topology optimization, and a growth simulator. However, it remains unclear to the reviewer how the machine learning model works in the framework. A more comprehensive and detailed exposition of the machine learning model's roles within the framework is necessary. Is the code going to be shared/made available?

(4) Can some discussion be devoted to the advantages of this paper's generative computational framework to other existing methods, since there has been some literature to modulate the mechanical response of architected materials or achieve lightweight architected materials (Nat Commun 14, 6630 (2023); Proceedings of the National Academy of Sciences, 119, 1, (2022)).

(5) What is the gap between the proposed framework and the real clinical applications? How can we employ it to general operation chains and what else we need to improve? It is suggested to make discussion on these aspects.

Reviewer #2:

Remarks to the Author:

This manuscript proposes an in silico framework to design 3D heterogeneous irregular microstructures, based on different building blocks optimized towards a target value of stress. It combines a machine learning model, a material property optimizer and a virtual growth simulator. In addition it applies the computational framework to the design of femur supports. The framework proposed is based on a previous work (Liu et al, Science 377, 2022) that already developed the virtual growth model and the construction of material databases. The main findings of this study are of interest and could be used to advance in the design of sophisticated irregular materials with a wide variety of applications such as orthopaedics. Nevertheless, there are areas that could benefit from refinement in order to enhance the overall clarity and precision of the manuscript.

- Why is firstly used the hydrostatic stress as the desired target value (section 2.1)? The value of 0.28 MPa needs further justification. Have you performed and checked the representative scenarios performed in section 2.1 with a target shear stress? The study afterwards presents the femur restoration as a potential application and uses in section 2.4 a target shear stress. Please elaborate.

- Figure 2a. Why a displacement loading of 1.5 mm is applied?

- Different target stress values are analyzed in section 2.2 (0.1, 0.3 and 0.5 MPa), which is very pertinent. However these values, are very similar to the value of 0.28 MPa used before. Have other orders of magnitude been analyzed?

- The target shear stress values of -0.25 and 0.25 MPa need further justification. In addition, during bone consolidation the stiffness of the callus will increase. How do the authors contemplate the modulation of the target shear stress level?
- The femur is modeled with a Young's modulus of 5 GPa, value that falls within the values of cortical and trabecular bones. However, the orthopaedic support is located within the diaphysis of the femur, which is composed of cortical bone and not trabecular bone. How is the femur modeled? Is the marrow cavity included in the model? How are the mechanical constraints of the load bearing defect considered?
- Which are the target values assumed to stimulate bone regeneration? If in vivo tests are contemplated to be performed other variables, such as porosity, pore size or specific surface area should be included in the optimization problem. Biological constraints are missing in the design of the support.

Reviewer #3:

Remarks to the Author:

This manuscript titled "Modulate Stress Distribution with Bio-Inspired Irregular Architected Materials Towards Optimal Tissue Support" presents an innovative approach to modulate stress distribution using bio-inspired irregular architected materials. A novel computational framework that combines a material database, machine learning models, macroscopic topology optimization, and a virtual growth simulator is established and verified. The results demonstrate the practical merit of this method, particularly in the context of orthopedic femur restoration. The figures and presentation within the manuscript are commendable and contribute to its overall clarity and impact. The approach outlined is innovative and could have substantial implications in the field. Therefore, I would recommend the publication of this work in Nature Communications after the authors address the following comments.

1. The manuscript lacks a clear, comprehensive description of the algorithm, which is the most important part of the paper. I have to carefully read "Materials and Methods" and even thoroughly read Supporting Information in order to understand the mechanism of the proposed algorithm. A high-level overview within the main body of the manuscript is necessary. The current description, particularly in Figure 1b-c, is vague and insufficient. A clearer presentation of the algorithm, possibly with a more illustrative figure or flowchart, would greatly enhance the manuscript's readability and scientific value. I would suggest moving some of the content in "Materials and Methods" to the manuscript.

2. In "Materials and Methods" section, the authors described the generation of specimens using a 40x40 array of building blocks. I am curious about the consistency of these 100 samples with identical frequencies. If there is significant variation, frequency alone might not adequately determine a specimen's mechanical properties. This concern casts doubt on the optimization of frequency without considering detailed distribution. This is the cornerstone of the whole algorithm. Thus, a deeper investigation and discussion on this aspect is needed.

3. During the topology optimization, my understanding is that frequency and density are defined on each building block or element. If this is true, how does the virtual growth algorithm accommodate spatially varying frequencies across the domain? For the specimen used of dataset, there is only one frequency defined on a 40 by 40 specimen. However, according to supplementary information, case 1 and 2 in the manuscript consist of 30 by 15 rectangular elements. If each rectangular element has unique frequency, how does virtual growth algorithm work. Are there an average or standardized approach to define these frequencies for the virtual growth algorithm. This aspect of the methodology requires further elaboration for clarity.

4. Given that the dataset is based on 40x40 specimens with constant frequency, questions arise about the neural network's applicability to scenarios with spatially varying frequencies or different specimen shapes. The manuscript should address whether the training on this dataset limits the model's generalizability. This is also a crucial cornerstone of the whole algorithm, if this assumption does not hold, then the whole algorithm will not work properly.

5. In the femur restoration example, the assumption of effective contact between the microstructure and femur needs scrutiny. Considering the challenges of aligning artificial materials with biological tissues, the potential stress concentration at the interface may adversely affect performance. Furthermore, since microstructure are used there will be stress concentration on the interface of the femur bone. How would this affect the stimulus effect of the microstructure supporting?

In conclusion, addressing these major comments will significantly strengthen the manuscript, offering clearer insights into the methodology and extending its applicability and robustness.

Response to Reviewers' comments

The authors are most grateful for the Reviewers' insightful comments and suggestions. We have thoroughly revised the main manuscript and Supplementary Information according to all the comments. The point-to-point response is highlighted in blue, and the corresponding changes in the revised manuscript and Supplementary Information are highlighted in red. Below we address each comment in detail.

Reviewer #1

The paper investigates the relationship between stress modulation and the irregularity of bio-inspired architected materials. It introduces a computational framework for optimizing the spatial distribution of basic building blocks to create irregular architected materials with heterogeneous disordered microstructures. The work includes a practical demonstration of this concept through 3D printed samples and highlights potential applications in orthopedic femur restoration. Overall, the reviewer is positive about the work - especially, the novel aspects regarding modulating stress distribution of large-scale irregular architected materials.

My comments are listed below:

Comment 1: It is unclear what criteria guided the author's selection of the four fundamental 3D building blocks. The paper would benefit from a discussion exploring alternative building blocks that could potentially enhance performance.

Response: The authors greatly appreciate the Reviewer for this valuable comment that gives us a chance to clarify our design philosophy. To address the reviewer's comment, we incorporate the following discussion in Sections 1.1 and 1.4 of the revised Supplementary Information.

"We require that the basic building blocks are representative and can be connected via rotation, ensuring the diversity and seamlessness of the generated materials. Specifically, we prescribe four types of basic building blocks, "cross", "arrow", "corner", and "line" shapes, in the two-dimensional (2D) case and four types of basic building blocks, "cross", "corner", "t", and "cross-line" shapes, in the three-dimensional (3D) case. These specified building blocks exhibit diverse geometries with each block capable of connecting only in specific directions to others, achieving a large material property space."

"To describe the material property space, we define the average Young's modulus and Poisson ratio by following the convention in [1]. Specifically, we define the average Young's modulus as

$$E^{\text{ave}} = \begin{cases} \frac{1}{2} \left(\frac{1}{\mathbb{S}_{1111}} + \frac{1}{\mathbb{S}_{2222}} \right) & \text{in 2D} \\ \frac{1}{3} \left(\frac{1}{\mathbb{S}_{1111}} + \frac{1}{\mathbb{S}_{2222}} + \frac{1}{\mathbb{S}_{3333}} \right) & \text{in 3D} \end{cases}$$

and the average Poisson's ratio as

$$\nu^{\text{ave}} = \begin{cases} -\frac{1}{2} \left(\frac{\mathbb{S}_{2211}}{\mathbb{S}_{1111}} + \frac{\mathbb{S}_{1122}}{\mathbb{S}_{2222}} \right) & \text{in 2D} \\ -\frac{1}{6} \left(\frac{\mathbb{S}_{2211}}{\mathbb{S}_{1111}} + \frac{\mathbb{S}_{1122}}{\mathbb{S}_{2222}} + \frac{\mathbb{S}_{3322}}{\mathbb{S}_{2222}} + \frac{\mathbb{S}_{2233}}{\mathbb{S}_{3333}} + \frac{\mathbb{S}_{3311}}{\mathbb{S}_{1111}} + \frac{\mathbb{S}_{1133}}{\mathbb{S}_{3333}} \right) & \text{in 3D} \end{cases}$$

where \mathbb{S} represents the fourth-order homogenized compliance tensor that can be derived from the matrix \mathbf{D} ."

"With the above definitions, we plot the material property space for both 2D and 3D scenarios in Figure 1. Here, the x -axis is the relative average Young's modulus defined as $E^{\text{ave}}/E^{\text{solid}}$, and E^{solid} is the Young's modulus of the solid material; the y -axis is the average Poisson's ratio, ν^{ave} . In Figure 1, the color wheels signify the frequency combinations present in the material database, and their positions correspond to $E^{\text{ave}}/E^{\text{solid}}-\nu^{\text{ave}}$ pairs averaged over the specimens of the same frequency combination. The shaded area in Figure 1 indicates the actual material properties achieved by all the specimens. The insets in both Figures 1a and b illustrate specimens generated based on their frequency combinations, respectively. According

to Figure 1, the 2D material property space spans $E^{\text{ave}}/E^{\text{solid}}$ values in $[0.001, 0.111]$ and ν^{ave} values in $[-0.309, 0.365]$; the 3D material property space spans the counterparts in $[0.003, 0.036]$ and $[0.008, 0.144]$, respectively. We remark again that the broad spectrum of material properties results from the selection of the representative building blocks.”

Figure 1: **Material property space in 2D and 3D cases.** The x -axis is the relative average Young’s modulus ($E^{\text{ave}}/E^{\text{solid}}$), and the y -axis is the average Poisson’s ratio (ν^{ave}). The color wheels represent the frequency combinations in the material database, and their positions represent the corresponding $E^{\text{ave}}/E^{\text{solid}}-\nu^{\text{ave}}$ pairs averaged over specimens of the same frequency combination, respectively. The shaded regions indicate the actual material properties achieved by all the specimens, and the insets illustrate the generated microstructures.

As for exploring alternative building blocks, we add the following discussion in Section 3 of the revised manuscript. “Additionally, our current focus lies on exploring a specific set of building blocks (in 2D and 3D, respectively) to modulate stress distribution. While these building blocks demonstrate satisfactory stress modulation effects, the performance can be potentially further improved by using alternative building blocks with different geometries because these geometries drastically affect the material property space (see Figures 4 and 6 in [1]). Furthermore, taking into account the anisotropy inherent in the homogenized material properties of the generated microstructures, we expect to further expand the material property space by permitting free elongation and rotation (as opposed to restricting it to multiples of $\pi/2$). This improvement requires the implementation of the virtual growth algorithm on an unstructured grid, which is our ongoing work.”

Comment 2: As shown in Figure 5f, the meaning of the ‘controlled element’ on the x -axis requires clarification. Furthermore, it is not clear to the reviewer why a significant stress disparity is observed among different controlled elements, while the initial state suggests a homogeneous distribution of building blocks.

Response: The authors are grateful for this comment and add the following clarifications in Section 2.4 of the revised manuscript. “To visualize the stress modulation effect, Figure 2f presents illustrative (left) and simulated (right) stress distributions. In the simulated stress distribution, the x -axis represents the finite elements in the stress control regions, while the y -axis represents the values of the selected stress measure (shear stress in this case). It is important to note that the initial stress distribution among the controlled elements is highly nonuniform due to the complex geometry of the femur (i.e., varying cross-sections and embedded fracture), despite the initial homogeneous distribution of building blocks within the support domain. However, after optimization, the actual stresses uniformly reach the target values.”

Figure 2: **Potential application of the irregular architected materials for orthopedic femur restoration.** **a**, A healthy femur efficiently distributes compressive forces, resulting in relatively uniform compressive stress distribution (red arrows in the inset). **b**, A fractured femur experiences concentrated compressive stress around the crack tip. **c**, Comparison between the traditional and proposed methods for orthopedic femur restoration. The traditional method induces stress shielding, while the proposed method stimulates growth with shear stress (green arrows) under the transmitted pressure (blue arrows). **d**, Simplified model of the proposed femur repair scheme with the applied displacement loading of U . Unit of target stresses, τ : MPa. **e**, The virtual growth process of the optimized 3D irregular architected materials. **f**, Illustrative (left) and simulated (right) stress distribution. In the right panel, the x -axis represents the finite elements in the stress control regions, and the y -axis represents the values of the selected stress measure (shear stress here). **g**, Comparison between the numerical and physical samples of optimized architected materials at varying scales.

Comment 3: The proposed framework integrates a machine learning model, topology optimization, and a

growth simulator. However, it remains unclear to the reviewer how the machine learning model works in the framework. A more comprehensive and detailed exposition of the machine learning model’s roles within the framework is necessary. Is the code going to be shared/made available?

Response: The authors thank the Reviewer for the great suggestion. We have added the following discussion and a new figure in Section 1 of the revised manuscript. “To generate the irregular architected materials, Figure 3 provides a detailed exposition of the proposed generative computational framework. As shown in Figure 3a, this framework begins by creating a discrete material database containing the prescribed building blocks, frequency combinations, and the generated microstructures. Following numerical homogenization [2], the material properties of the microstructures are derived, and these properties are then correlated with the microstructures’ frequency combinations. To establish a continuous relationship between the frequency combination and the material properties, a machine learning model is trained (Figure 3b) to predict the independent components of the stiffness matrix based on the input frequency combination. Subsequently, macroscopic topology optimization is performed (Figure 3c) to optimize the design variables defined on the finite elements within the design domain to meet the target. During this optimization process, a density variable is defined to describe whether a finite element is filled with a microstructure (solid) or not (void). Additionally, frequency variables are defined to characterize the frequency combination of building blocks if the finite element is filled with a microstructure. After defining these design variables, the machine learning model from Figure 3b is utilized to predict the spatial distribution of material properties. Subsequently, finite element analysis (FEA) is conducted to derive the stress distribution. Guided by the gradient information, the design variables undergo iterative updates by the optimizer until the actual stress distribution converges to the target. Finally, based on the optimized density and frequency variables, we apply the virtual growth algorithm to “grow” the irregular architected materials capable of modulating the stress distribution. Here, we remark that the original virtual growth algorithm in [1] mainly yields materials with one intuitively specified frequency combination (i.e., one finite element and one microstructure). In the current study, we generalize this algorithm to account for spatially varying and optimized frequency combinations and densities (solid or void) within different finite elements. In the generalized algorithm (Figure 3d), individual finite elements are assigned unique frequency combinations, and all building blocks within a finite element adhere to the corresponding frequency combination of that finite element. Consequently, the generated materials consist of heterogeneous microstructures (i.e., spatially varying frequencies across the domain), and the building blocks within each microstructure conform to the corresponding frequency combination while maintaining seamless integration between neighboring microstructures.”

Figure 3: **Proposed generative computational framework.** **a**, Material database creation. The bottom panel is the material property space, and the details are explained in Section 1.4 of the Supplementary Information. **b**, Machine learning (ML) model for predicting the material properties based on the input frequency combination. **c**, Macroscopic topology optimization that determines the optimal design variables to modulate the stress distribution to the target. **d**, Virtual growth simulator for generating the irregular architected materials based on the optimized design variables.

For code availability, the computer code is available from the corresponding authors upon request, and we have plans to make the code fully available in the future. In addition, all the data that support the main finding of the paper will be accessible on GitHub.

Comment 4: Can some discussion be devoted to the advantages of this paper’s generative computational framework to other existing methods, since there has been some literature to modulate the mechanical response of architected materials or achieve lightweight architected materials (Nat Commun 14, 6630 (2023); Proceedings of the National Academy of Sciences, 119, 1, (2022)).

Response: The authors thank the Reviewer for the great suggestion. We have added the following discussion in Section 1 of the revised manuscript. “In comparison to related existing research [3, 4] that primarily focus on designing deterministic and nearly periodic architected materials, the present study introduces a generative computational framework for crafting stochastic and aperiodic materials. The incorporation of stochasticity and aperiodicity holds the potential to enhance failure resistance [5]. Furthermore, unlike the existing studies [3, 4] that predominantly investigate the performance (such as stiffness) of individual microstructures, our current study aims at modulating the global stress distribution of functional material pieces comprising heterogeneous microstructures.”

Comment 5: What is the gap between the proposed framework and the real clinical applications? How can we employ it to general operation chains and what else we need to improve? It is suggested to make discussion on these aspects.

Response: The authors are grateful for this insightful comment and have added the following discussion in Section 3 of the revised manuscript.

“Moving forward, we anticipate incorporating more biological considerations into the framework and conducting in-vivo tests to validate the designed support for orthopedic femur restoration. To achieve this goal, we plan to employ a more representative femur model reflecting its porous structures and spatially varying material properties. In addition, we need to incorporate biological design variables [6] including the porosity, pore size, and the specific surface area of the support and the biological constraints including the minimum stiffness and fracture resistance [7] of the femur. We also need to ensure the biocompatibility of the support material [8]. Furthermore, during femur restoration, as the bone consolidates and the callus stiffness increases, enlarging the target shear stress may be necessary to maintain the same level of micromovement of femur fragments around the fracture. To accommodate varying target stress levels, we can reformulate the design task into a simultaneous control problem akin to Figure 5 (of the revised manuscript). Consequently, the generated support can modulate stress to different levels as the femur progresses through different healing states.”

“Beyond biological considerations, the stress modulation performance can be further enhanced by exploring alternative building blocks. In our current study, we concentrate on investigating a specific set of building blocks (in 2D and 3D, respectively) to modulate stress distribution. Referring to Figures 4 and 6 in [1], the use of alternative building blocks with different geometries can significantly impact the material property space and, consequently, influence stress modulation effects. Additionally, due to the anisotropy of material properties in microstructures, expanding the material property space by allowing free elongation and rotation (as opposed to restricting it to multiples of $\pi/2$) is a potential improvement.”

Reviewer #2

This manuscript proposes an in silico framework to design 3D heterogeneous irregular microstructures, based on different building blocks optimized towards a target value of stress. It combines a machine learning model, a material property optimizer and a virtual growth simulator. In addition it applies the computational framework to the design of femur supports. The framework proposed is based on a previous work (Liu et al, Science 377, 2022) that already developed the virtual growth model and the construction of material databases. The main findings of this study are of interest and could be used to advance in the design of sophisticated irregular materials with a wide variety of applications such as orthopaedics. Nevertheless, there are areas that could benefit from refinement in order to enhance the overall clarity and precision of the manuscript.

Comment 1: Why is firstly used the hydrostatic stress as the desired target value (section 2.1)? The value of 0.28 MPa needs further justification. Have you performed and checked the representative scenarios performed in section 2.1 with a target shear stress? The study afterwards presents the femur restoration as a potential application and uses in section 2.4 a target shear stress. Please elaborate.

Response: The authors are grateful for the Reviewer’s comment and have added the following investigations and explanations in Section 3.4 of the revised Supplementary Information and Section 2.1 of the revised manuscript

“In the main manuscript, we modulate the hydrostatic stress to a target value of $\bar{\sigma}^h = 0.28$ MPa under an applied displacement of $u = 1.5$ mm. The selection of this setup is solely for demonstration purposes, and the proposed computational framework is not confined to a specific stress measure, target value, or applied displacement. To illustrate the versatility of the framework, we take Case 2 as an example and explore stress modulation effects under varying setups. Figure 4 displays the initial, optimized, and target stresses, with the insets illustrating the design setup. Specifically, in Figure 4a, we modulate the shear stress (instead of hydrostatic stress) to two target values, $\bar{\tau} = 0.02$ MPa and $\bar{\tau} = 0.04$ MPa, respectively. In Figure 4b, we explore two different target values for hydrostatic stress, $\bar{\sigma}^h = 0.18$ MPa and $\bar{\sigma}^h = 0.38$ MPa (as opposed to $\bar{\sigma}^h = 0.28$ MPa). Finally, in Figure 4c, we consider two different applied displacements, $u = 0.20$ mm and $u = 0.25$ mm (instead of $u = 0.15$ mm). Table 1 summarizes the design setups and the corresponding relative stress modulation errors. According to Figure 4 and Table 1, the stress distribution aligns with the target after optimization across diverse setups, indicating the robust generality of the proposed framework for stress modulation.”

Figure 4: **Investigation of stress modulation effects under varying setups.** **a**, The stress measure is changed from hydrostatic to shear stress. **b**, The target value is changed from 0.28 MPa to 0.18 and 0.38 MPa. **c**, The applied displacement is changed from 0.15 mm to 0.20 and 0.25 mm. The insets illustrate the updated design setups.

Table 1: **Summary of designs setups and relative stress errors in Figure 4**

Stress measure	Target value (MPa)	Applied displacement (mm)	Loading direction	Initial stress error	Optimized stress error
Shear ($\bar{\tau}$)	0.02	0.15	Upward	532.0%	2.8%
Shear ($\bar{\tau}$)	0.04	0.15	Upward	690.0%	2.6%
Hydrostatic ($\bar{\sigma}^h$)	0.18	0.15	Rightward	660.0%	3.2%
Hydrostatic ($\bar{\sigma}^h$)	0.38	0.15	Rightward	759.0%	4.0%
Hydrostatic ($\bar{\sigma}^h$)	0.28	0.20	Rightward	687.0%	3.2%
Hydrostatic ($\bar{\sigma}^h$)	0.28	0.25	Rightward	647.0%	3.2%

Comment 2: Figure 2a. Why a displacement loading of 1.5 mm is applied?

Response: The authors thank the reviewer for raising this point. For demonstration purposes, we would like to choose a displacement load that is clearly visible by the DIC system and, at the same time, stays in a relatively small deformation range. Thus, 1.5mm is the result of the choice. This choice ensures adherence to the linear elasticity assumption and mitigates the risk of potential fractures. The proposed generative computational framework is not restricted to a specific applied displacement. As part of our exploration, we also consider alternative applied displacements, namely $u = 0.20$ mm and $u = 0.25$ mm, as shown in Figure 4c (within this document and in Section 3.4 of the revised SI) and detailed in the last two rows of Table 1 (within this document and in Section 3.4 of the revised SI). The results demonstrate consistently small relative stress modulation errors across varying applied displacements.

Comment 3: Different target stress values are analyzed in section 2.2 (0.1, 0.3 and 0.5 MPa), which is very pertinent. However these values, are very similar to the value of 0.28 MPa used before. Have other orders of magnitude been analyzed?

Response: The authors thank the Reviewer for this insightful comment and have added the following investigation in Section 2.2 of the revised manuscript and Section 3.4 of the revised Supplementary Information.

“In the main manuscript, we set the target stress values as 0.1, 0.3, and 0.5 MPa for distinct control regions. Here, we explore the stress modulation effects for other magnitudes of target values. In Figure 5a, we consider 0.01, 0.03, and 0.05 MPa as the target values, and all the other setups remain unchanged. Likewise, we consider 0.2, 0.6, and 1.0 MPa as the target values in Figure 5b. We observe that the actual stress distribution closely aligns with the target for both scenarios, demonstrating the robust stress modulation capability of the proposed framework.”

Figure 5: **Stress modulation effects for different magnitudes of target values.** **a**, The target stress values are 0.01, 0.03, and 0.05 MPa. **b**, The target stress values are 0.2, 0.6, and 1.0 MPa.

Comment 4: The target shear stress values of -0.25 and 0.25 MPa need further justification. In addition, during bone consolidation the stiffness of the callus will increase. How do the authors contemplate the modulation of the target shear stress level?

Response: The authors thank the Reviewer for raising these important points. In terms of the target stress values, we have incorporated more representative targets, namely -5.0 and 5.0 MPa, for the two stress control regions and also added the following discussion in Section 2.4 of the revised manuscript and Section 3.4 of the revised Supporting Information.

“In this work, for illustrative purposes, we designate the target shear stresses as $\bar{\tau} = -5.0$ and $\bar{\tau} = 5.0$ MPa for control regions 1 and 2 (the translucent view in Figure 2d), respectively. These specified target values are intentionally well below the shear strength range of human femurs (51.6 MPa–65.3 MPa for cortical bone [9]) to mitigate the risk of inducing additional fractures. Notably, these target stress values correspond

to a relative micromotion of 0.3 mm of the femur around the fracture (see Figure 8 6), falling within the suggested range of [0.2, 1.0] mm for femur restoration [10].”

“Here, we present both the mesh discretization and the displacement field in the z direction after optimization in Figure 6. Upon close examination of the displacement field in the zoomed-in view, we observe that the relative displacement of the two femur fragments perpendicular to the fracture is approximately 0.3 mm. This value falls within the range of [0.2, 1.0] mm, suggesting the potential to facilitate femur restoration [10].”

Figure 6: **Finite element discretization and the optimized displacement field in z -direction of orthopedic femur restoration.** The model consists of the femur embedded with a pre-fracture and artificial support. Both components are discretized with tetrahedral elements due to complex geometries. In the visualization, color signifies the optimized displacement field in the z -direction, with the inset offering a zoom-in view of the displacement field around the fracture.

For the bone consolidation point, we have included the following discussion in Section 3 of the revised manuscript. “During femur restoration, as the bone consolidates and the callus stiffness increases, enlarging the target shear stress may be necessary to maintain the same level of micromovement of femur fragments around the fracture. To accommodate varying target stress levels, we can reformulate the design task into a simultaneous control problem akin to Figure 5 of the revised manuscript. Consequently, the generated support can modulate stress to different levels as the femur progresses through different healing states.”

Comment 5: The femur is modeled with a Young’s modulus of 5 GPa, value that falls within the values of cortical and trabecular bones. However, the orthopaedic support is located within the diaphysis of the femur, which is composed of cortical bone and not trabecular bone. How is the femur modeled? Is the marrow cavity included in the model? How are the mechanical constraints of the load bearing defect considered?

Response: The authors thank the Reviewer for raising this important point. We have added the following descriptions in Sections 2.4 and 3 of the revised manuscript.

“Simplifying our femur model, we treat it as an isotropic passive material with Young’s modulus of $E = 5.0$ GPa. This selection is informed by the predominant cortical bone composition in the femur’s diaphysis, where Young’s modulus is approximately 16.7 GPa [11]. In addition, the homogenized Young’s modulus of the femur should be further reduced to account for the marrow cavity within the femur whose Young’s modulus ranges from 0.25 to 24.7 kPa [12]. Furthermore, the load-bearing defect (fracture) is directly reflected in the femur geometries (see Figure 6). Here, we note that the isotropy assumption yields results comparable to those obtained with orthotropic materials [13], and the proposed framework is applicable for

a more intricate femur modeling with additional computational cost.”

“Moving forward, we anticipate incorporating more biological considerations into the framework and conducting in-vivo tests to validate the designed support for orthopedic femur restoration. To achieve this goal, we plan to employ a more representative femur model reflecting its porous structures and spatially varying material properties.”

Comment 6: Which are the target values assumed to stimulate bone regeneration? If in vivo tests are contemplated to be performed other variables, such as porosity, pore size or specific surface area should be included in the optimization problem. Biological constraints are missing in the design of the support.

Response: The authors are grateful for the Reviewer’s comment. As for the target stress values, we add the following descriptions in Section 2.4 of the revised manuscript. “In this work, for illustrative purposes, we designate the target shear stresses as $\bar{\tau} = -5.0$ and $\bar{\tau} = 5.0$ MPa for control regions 1 and 2 (the translucent view in Figure 2d), respectively. These specified target values are intentionally well below the shear strength range of human femurs (51.6 MPa–65.3 MPa for cortical bone [9]) to mitigate the risk of inducing additional fractures. Notably, these target stress values correspond to a relative micromotion of 0.3 mm of the femur around the fracture (see Figure 6), falling within the suggested range of [0.2, 1.0] mm for femur restoration [10].”

Regarding the biological considerations, we add the following discussion in Section 3 of the revised manuscript. “If in-vivo tests are performed in the future, we need to incorporate biological design variables [6] including the porosity, pore size, and the specific surface area of the support and the biological constraints including the minimum stiffness and fracture resistance [7] of the femur. We also need to ensure the biocompatibility of the support material [8].”

Reviewer #3

This manuscript titled “Modulate Stress Distribution with Bio-Inspired Irregular Architected Materials Towards Optimal Tissue Support” presents an innovative approach to modulate stress distribution using bio-inspired irregular architected materials. A novel computational framework that combines a material database, machine learning models, macroscopic topology optimization, and a virtual growth simulator is established and verified. The results demonstrate the practical merit of this method, particularly in the context of orthopedic femur restoration. The figures and presentation within the manuscript are commendable and contribute to its overall clarity and impact. The approach outlined is innovative and could have substantial implications in the field. Therefore, I would recommend the publication of this work in Nature Communications after the authors address the following comments.

Comment 1: The manuscript lacks a clear, comprehensive description of the algorithm, which is the most important part of the paper. I have to carefully read “Materials and Methods” and even thoroughly read Supplementary Information in order to understand the mechanism of the proposed algorithm. A high-level overview within the main body of the manuscript is necessary. The current description, particularly in Figure 1b-c, is vague and insufficient. A clearer presentation of the algorithm, possibly with a more illustrative figure or flowchart, would greatly enhance the manuscript’s readability and scientific value. I would suggest moving some of the content in “Materials and Methods” to the manuscript.

Response: The authors greatly appreciate the Reviewer’s valuable comment. Following the suggestion, we have added a new figure and the following contents to Section 1 of the revised manuscript.

“To generate the irregular architected materials, Figure 3 provides a detailed exposition of the proposed generative computational framework. As shown in Figure 3a, this framework begins by creating a discrete material database containing the prescribed building blocks, frequency combinations, and the generated microstructures. Following numerical homogenization [2], the material properties of the microstructures are derived, and these properties are then correlated with the microstructures’ frequency combinations. To establish a continuous relationship between the frequency combination and the material properties, a machine learning model is trained (Figure 3b) to predict the independent components of the stiffness matrix based on the input frequency combination. Subsequently, macroscopic topology optimization is performed (Figure 3c) to optimize the design variables defined on the finite elements within the design domain to meet the target. During this optimization process, a density variable is defined to describe whether a finite

element is filled with a microstructure (solid) or not (void). Additionally, frequency variables are defined to characterize the frequency combination of building blocks if the finite element is filled with a microstructure. After defining these design variables, the machine learning model from Figure 3b is utilized to predict the spatial distribution of material properties. Subsequently, finite element analysis (FEA) is conducted to derive the stress distribution. Guided by the gradient information, the design variables undergo iterative updates by the optimizer until the actual stress distribution converges to the target. Finally, based on the optimized density and frequency variables, we apply the virtual growth algorithm to “grow” the irregular architected materials capable of modulating the stress distribution. Here, we remark that the original virtual growth algorithm in [1] mainly yields materials with one intuitively specified frequency combination (i.e., one finite element and one microstructure). In the current study, we generalize this algorithm to account for spatially varying and optimized frequency combinations and densities (solid or void) within different finite elements. In the generalized algorithm (Figure 3d), individual finite elements are assigned unique frequency combinations, and all building blocks within a finite element adhere to the corresponding frequency combination of that finite element. Consequently, the generated materials consist of heterogeneous microstructures (i.e., spatially varying frequencies across the domain), and the building blocks within each microstructure conform to the corresponding frequency combination while maintaining seamless integration between neighboring microstructures.”

Comment 2: In “Materials and Methods” section, the authors described the generation of specimens using a 40x40 array of building blocks. I am curious about the consistency of these 100 samples with identical frequencies. If there is significant variation, frequency alone might not adequately determine a specimen’s mechanical properties. This concern casts doubt on the optimization of frequency without considering detailed distribution. This is the cornerstone of the whole algorithm. Thus, a deeper investigation and discussion on this aspect is needed.

Response: The authors thank the Reviewer for this insightful comment. We have added the following investigation and discussion in Section 1.3 of the revised Supplementary Information. “Considering the stochastic nature of the virtual growth algorithm, the material properties in the generated database exhibit randomness. Here, we study the material property distribution of the created specimens. Figure 7 displays the distribution of the D_{11} component of the homogenized elastic modulus (\mathbf{D}) for 100 specimens corresponding to randomly selected 9 frequency combinations. In Figure 7, the x -axis represents the material property (D_{11} in this case). The left y -axis denotes the statistical frequency (count), while the right y -axis represents the probability density function. The inset in each panel presents the associated frequency combination of building blocks. Notably, the material properties of the generated specimens generally exhibit unimodality and near-zero skewness. Therefore, we assume that the material properties follow a normal distribution with their mean values determined by the frequency combinations of building blocks.”

Figure 7: **Material property distribution of the created specimens.** The x -axis is the material property (D_{11} component of the \mathbf{D} matrix here), the left y -axis is the statistical frequency, and right y -axis is the probability density function. The inset color wheels represent the frequency combinations of building blocks.

In the current study, we modulate the stress distribution in an average sense, and therefore the frequency combinations (and the densities) are sufficient to optimize the irregular architected materials through the mean values of material properties. Despite the potential deviations in the modulated stress, the stress convergence studies in Figure 8e (Figure 3e of the revised manuscript) and Figure 9 (Figure 10 of the revised Supplementary Information) suggest that such deviations diminish as the number of building blocks in each microstructure increases. In addition, the average stress distribution eventually converges to the target values.

Figure 8: **Manipulating mechanical stress distribution in varied geometric regions.** **a**, Design domain, boundary conditions, and three distinct stress control regions (rectangle, square, and ring-shape in Cases 1–3, respectively). The variable $u = 1.5$ mm represents displacement loading. Variables σ^h and $\bar{\sigma}^h$ are actual and target hydrostatic stresses (in MPa), respectively. **b**, Optimized irregular architected materials made of randomly yet optimally distributed microstructures. **c**, Spatially varying distribution of properties, as represented by the D_{11} elastic modulus (in MPa). **d**, Precise stress manipulation (in MPa) is realized by the spatially varying material property. **e**, Stress convergence study of the generated material in Case 3. The variable k represents the number of basic building blocks in one direction within one microstructure. Each error bar represents the distribution of hydrostatic stress of one specimen. The dots and the half-lengths of error bars indicate the mean values and the standard deviations, respectively. **f**, Experimental setup for measuring the displacement field. **g–h**, Displacements (in mm) in the loading direction obtained experimentally from the digital image correlation (DIC) and numerically from the finite element analysis (FEA), respectively. **i–k**, Detailed comparisons of the average displacements within the control regions for the 3 cases. The measured values and error bars of experiments are also plotted.

Figure 9: **Stress convergence studies for mechanical stress modulation within distinct control regions.** **a**, Case 1; **b**, Case 2. The plots illustrate the relationship between the hydrostatic stress measure, denoted by σ^h (in MPa), and the number of basic building blocks in one direction within one microstructure. Each error bar represents one numerical sample. The circular dot of the error bar represents the mean hydrostatic stress in the control region, and the half length of the error bar represents the standard deviation.

Comment 3: During the topology optimization, my understanding is that frequency and density are defined on each building block or element. If this is true, how does the virtual growth algorithm accommodate spatially varying frequencies across the domain? For the specimen used of dataset, there is only one frequency defined on a 40 by 40 specimen. However, according to supplementary information, case 1 and 2 in the manuscript consist of 30 by 15 rectangular elements. If each rectangular element has unique frequency, how does virtual growth algorithm work. Are there an average or standardized approach to define these frequencies for the virtual growth algorithm. This aspect of the methodology requires further elaboration for clarity.

Response: The authors are grateful for the Reviewer’s comment, and in response, we add the following descriptions and a new figure in Section 1 of the revised manuscript. “Here, we remark that the original virtual growth algorithm in [1] mainly yields materials with one intuitively specified frequency combination (i.e., one finite element and one microstructure). In the current study, we generalize this algorithm to account for spatially varying and optimized frequency combinations and densities (solid or void) within

different finite elements. In the generalized algorithm (Figure 3d), individual finite elements are assigned unique frequency combinations, and all building blocks within a finite element adhere to the corresponding frequency combination of that finite element. Consequently, the generated materials consist of heterogeneous microstructures (i.e., spatially varying frequencies across the domain), and the building blocks within each microstructure conform to the corresponding frequency combination while maintaining seamless integration between neighboring microstructures.”

Comment 4: Given that the dataset is based on 40x40 specimens with constant frequency, questions arise about the neural network’s applicability to scenarios with spatially varying frequencies or different specimen shapes. The manuscript should address whether the training on this dataset limits the model’s generalizability. This is also a crucial cornerstone of the whole algorithm, if this assumption does not hold, then the whole algorithm will not work properly.

Response: The authors thank the Reviewer for this insightful comment and agree that the generality of the trained neural network is the foundation of the proposed framework. In the revised manuscript and Supplementary Information, we have performed the stress convergence studies for all three cases from Figure 3 of the revised manuscript. In Figure 8e (Figure 3e of the revised manuscript) and Figure 9 (Figure 10 of the revised Supplementary Information), we explore the stress modulation effects across different microstructure shapes, ranging from 1×1 to 12×12 building blocks. Importantly, the frequency combinations vary spatially within distinct finite elements. Figures 8e and 9 reveal that the mean value of the modulated stress converges to the target, while the standard deviation diminishes as the microstructure shape approaches the one used for training the neural network (40×40). This observation highlights the generality of the trained neural network, demonstrating its accuracy in predicting material properties for scenarios involving spatially varying frequency combinations and diverse microstructure shapes.

Comment 5: In the femur restoration example, the assumption of effective contact between the microstructure and femur needs scrutiny. Considering the challenges of aligning artificial materials with biological tissues, the potential stress concentration at the interface may adversely affect performance. Furthermore, since microstructure are used there will be stress concentration on the interface of the femur bone. How would this affect the stimulus effect of the microstructure supporting?

In conclusion, addressing these major comments will significantly strengthen the manuscript, offering clearer insights into the methodology and extending its applicability and robustness.

Response: The authors are grateful for the Reviewer’s important comment. We have added the following discussion in Section 3 of the revised manuscript. “In the context of orthopedic femur restoration, our assumption entails a perfect bounding interface between the femur and the support. It is crucial to note that in real in-vivo applications, aligning artificial materials with biological tissues demands a tailored design of interface and careful fabrication. Additionally, there is a possibility of local stress concentration along the interface between the femur and the support composed of artificial microstructures. Despite the potential for stress concentration, the proposed generative computational framework primarily modulates the homogenized stress distribution rather than focusing on local stress patterns. This approach allows for the control of the global relative displacement of the femur perpendicular to the fracture, as illustrated in Figure 6, with the ultimate goal of stimulating tissue regeneration. We also note that the proposed solution approaches the homogenized limit, when employing a greater number of building blocks in each finite element, characterized by smaller variations of local responses (Figures 8e and 9).”

References

- [1] K. Liu, R. Sun, C. Daraio, Growth rules for irregular architected materials with programmable properties, *Science* 377 (6609) (2022) 975–981. doi:10.1126/science.abn1459.
- [2] A. Vigliotti, D. Pasini, Stiffness and strength of tridimensional periodic lattices, *Computer Methods in Applied Mechanics and Engineering* 229–232 (2012) 27–43. doi:10.1016/j.cma.2012.03.018.

- [3] J.-H. Bastek, S. Kumar, B. Telgen, R. N. Glaesener, D. M. Kochmann, Inverting the structure–property map of truss metamaterials by deep learning, *Proceedings of the National Academy of Sciences* 119 (1) (2022) e2111505119. doi:10.1073/pnas.2111505119.
- [4] B. Peng, Y. Wei, Y. Qin, J. Dai, Y. Li, A. Liu, Y. Tian, L. Han, Y. Zheng, P. Wen, Machine learning-enabled constrained multi-objective design of architected materials, *Nature Communications* 14 (1) (2023) 6630. doi:10.1038/s41467-023-42415-y.
- [5] M. Zaiser, S. Zapperi, Disordered mechanical metamaterials, *Nature Reviews Physics* 5 (11) (2023) 679–688. doi:10.1038/s42254-023-00639-3.
- [6] S. Hollister, R. Maddox, J. Taboas, Optimal design and fabrication of scaffolds to mimic tissue properties and satisfy biological constraints, *Biomaterials* 23 (20) (2002) 4095–4103. doi:10.1016/S0142-9612(02)00148-5.
- [7] Y. Jia, O. Lopez-Pamies, X. S. Zhang, Controlling the fracture response of structures via topology optimization: From delaying fracture nucleation to maximizing toughness, *Journal of the Mechanics and Physics of Solids* 173 (2023) 105227. doi:10.1016/j.jmps.2023.105227.
- [8] S. J. Hollister, Porous scaffold design for tissue engineering, *Nature Materials* 4 (7) (2005) 518–524. doi:10.1038/nmat1421.
- [9] C. Turner, T. Wang, D. Burr, Shear Strength and Fatigue Properties of Human Cortical Bone Determined from Pure Shear Tests, *Calcified tissue international* 69 (2002) 373–8. doi:10.1007/s00223-001-1006-1.
- [10] H. Ebrahimi, M. Rabinovich, V. Vuleta, D. Zalcman, S. Shah, A. Dubov, K. Roy, F. S. Siddiqui, E. H. Schemitsch, H. Bougherara, R. Zdero, Biomechanical properties of an intact, injured, repaired, and healed femur: An experimental and computational study, *Journal of the Mechanical Behavior of Biomedical Materials* 16 (2012) 121–135. doi:10.1016/j.jmbbm.2012.09.005.
- [11] S. Arabnejad, B. Johnston, M. Tanzer, D. Pasini, Fully porous 3D printed titanium femoral stem to reduce stress-shielding following total hip arthroplasty, *Journal of Orthopaedic Research* 35 (8) (2017) 1774–1783. doi:10.1002/jor.23445.
- [12] L. E. Jansen, N. P. Birch, J. D. Schiffman, A. J. Crosby, S. R. Peyton, Mechanics of intact bone marrow, *Journal of the Mechanical Behavior of Biomedical Materials* 50 (2015) 299–307. doi:10.1016/j.jmbbm.2015.06.023.
- [13] V. Baca, Z. Horak, P. Mikulenka, V. Dzupa, Comparison of an inhomogeneous orthotropic and isotropic material models used for FE analyses, *Medical Engineering & Physics* 30 (7) (2008) 924–930. doi:10.1016/j.medengphy.2007.12.009.

Reviewers' Comments:

Reviewer #1:

Remarks to the Author:

The authors have adequately responded to the concerns raised. I recommend this paper for publication.

Reviewer #2:

Remarks to the Author:

The authors have addressed all my comments and the paper is now suitable for publication.

Reviewer #3:

Remarks to the Author:

The authors have address all of my comments. I do not have further questions and would recommend the publication of the article